

# TGF-beta signalling in bovine mammary gland involution and a comparative assessment of MAC-T and BME-UV1 cells as in vitro models for its study

Charlotte Alexandra Mitz and Alicia Mercedes Viloria-Petit

Department of Biomedical Sciences, Ontario Veterinary College, University of Guelph, Guelph, ON, Canada

## ABSTRACT

The goal of the dairy industry is ultimately to increase lactation persistency, which is the length of time during which peak milk yield is sustained. Lactation persistency is determined by the balance of cell apoptosis and cell proliferation; when the balance is skewed toward the latter, this results in greater persistency. Thus, we can potentially increase milk production in dairy cows through manipulating apoptogenic and antiproliferative cellular signaling that occurs in the bovine mammary gland. Transforming growth factor beta 1 (TGFβ1) is an antiproliferative and apoptogenic cytokine that is upregulated during bovine mammary gland involution. Here, we discuss possible applications of TGFβ1 signaling for the purposes of increasing lactation persistency. We also compare the features of mammary alveolar cells expressing SV-40 large T antigen (MAC-T) and bovine mammary epithelial cells-clone UV1 (BME-UV1) cells, two extensively used bovine mammary epithelial cell lines, to assess their appropriateness for the study of TGFβ1 signaling. TGFβ1 induces apoptosis and arrests cell growth in BME-UV1 cells, and this was reported to involve suppression of the somatotropic axis. Conversely, there is no proof that exogenous TGFβ1 induces apoptosis of MAC-T cells. In addition to TGFβ1's different effects on apoptosis in these cell lines, hormones and growth factors have distinct effects on TGFβ1 secretion and synthesis in MAC-T and BME-UV1 cells as well. MAC-T and BME-UV1 cells may behave differently in response to TGFβ1 due to their contrasting phenotypes; MAC-T cells have a profile indicative of both myoepithelial and luminal populations, while the BME-UV1 cells exclusively contain a luminal-like profile. Depending on the nature of the research question, the use of these cell lines as models to study TGFβ1 signaling should be carefully tailored to the questions asked.

Corresponding author
Alicia Mercedes Viloria-Petit,
aviloria@uoguelph.ca

## INTRODUCTION

The dairy industry could gain significant financial advantage by extending the persistency of lactation (*Capuco et al., 2003*), which is defined as the number of days in which a

constant milk yield is maintained (*Grossman, Hartz & Koops, 1999*). One of the ways in which dairy farmers increase milk yield is through exogenous administration of recombinant bovine growth hormone (rBGH), which is also known as recombinant bovine somatotropin (rBST) (*Etherton & Bauman, 1998*). While the use of this hormone presents no health concerns to humans (*MacLeod et al., 1999*), it is costly to the health of dairy cows (*Dohoo et al., 2003*). To assess the impact of rBGH/rBST on animal health, a meta-analysis was conducted by the Canadian Veterinary Medical Association. The report stated that administration of rBGH/rBST to dairy cows increased the risk of clinical mastitis by up to 25%, of infertility by 40%, and of lameness by up to 55% (*Dohoo et al., 2003*). The use of rBGH/rBST has been banned in Canada, however, this hormone is commonly used on dairy cows in the US (*Forge, 1998*; *Sechen, 2013*). BGH/BST is considered the predominant galactopoietic hormone in dairy cows (*Tucker, 2000*) and based on studies in rodents (*Collier, Annen-Dawson & Pezeshki, 2012*) and in bovine mammary gland explants (*Palin, Farmer & Duarte, 2017*), BGH/BST acts in conjunction with the well-documented galactopoietic hormone, prolactin (PRL). Many of the galactopoietic effects of BGH/BST are thought to be mediated through insulin-like growth factor I (IGF-I; *Bauman, 1999*). Binding of IGF-I to the IGF receptor (IGFR) activates IGFR intrinsic tyrosine kinase activity, and leads to the activation of the phosphatidylinositol-4,5-biphosphate 3 kinase (PI3K)/Akt (*Argetsinger et al., 1993*). In this review, we will refer to this BGH/IGF-1/PI3K/Akt signaling pathway as the somatotropic axis. One of the ways that IGF-I enhances survival is through suppression of the cytokine transforming growth factor beta 1 (TGFβ1; *Gajewska & Motyl, 2004*). MAC-T (mammary alveolar cells expressing SV-40 large T antigen) and BME-UV1 (bovine mammary epithelial cells-clone UV1) are two of the most popular immortalized bovine mammary epithelial cell lines used in the study of mammalian lactation (*Jedrzejczak & Szatkowska, 2014*). TGFβ1 induces apoptosis and arrests cell growth of BME-UV1 cells, and this was reported to involve suppression of the somatotropic axis (*Kolek et al., 2003*; *Gajewska & Motyl, 2004*). Interestingly, there is no literature to support that exogenous TGFβ1 induces apoptosis of MAC-T cells, nor suppresses the somatotropic pathway. In this article, we present an overview of TGFβ signaling in the bovine mammary gland, we review and compare the literature on the MACT-T and BME-UV1 cell lines, and we provide possible explanations as to why different responses to TGFβ1 are observed. We also provide suggestions along the way that might aid in improving our understanding of TGFβ function in the bovine mammary gland, as well as potential means to alter TGFβ signaling as a way of increasing milk production.

## Survey methodology

Relevant journal articles were obtained by performing a search for "MAC-T TGFbeta," "BME-UV1 TGFbeta," "Bovine mammary TGFbeta," "Bovine mammary epithelial TGFbeta," and "Bovine mammary stroma TGFbeta" on the databases Web of Science, Google Scholar, PubMed, and JSTOR.

## TGFβ1 in the bovine mammary gland

Lactation is followed by involution, which is the process by which the mammary gland transitions from a lactating to a non-lactating state (*Hurley, 1989*), and is also the initiation of the dry period. The dry period is a term used in the dairy industry and is defined as the period between cessation of milk removal at dry-off and the initiation of milking at the subsequent calving (*Hurley & Loor, 2011*). In bovine mammary gland involution, there is a decline in systemic hormone (e.g., BGH/BST and PRL) levels, and this is thought to initiate apoptosis of senescent secretory epithelial cells (*Pai & Horseman, 2011*). These secretory epithelial cells surround the lumen of the alveoli connected to a basement membrane, and are themselves surrounded by a layer of contractile myoepithelial cells (*Nickerson & Akers, 2011*). During lactation, secretory epithelial cells are responsible for synthesizing and secreting milk into the lumen, which is then drained into the ducts and collected by the lactiferous sinus before milk removal (*Nickerson & Akers, 2011*). The role of myoepithelial cells is to contract to allow the secretory epithelial cells to secrete milk out of the lumen and into the ducts (*Bruckmaier & Blum, 1998*). Myoepithelial cells are also important for the establishment of apical-basal polarity and for cellular survival, as they secrete components of the basement membrane, which by binding integrins on the basal site of luminal epithelial cells, activate cellular signaling that direct these events (*Weaver et al., 2002*).

Transforming growth factor beta (TGFβ) belongs to a family of growth factors of at least 33 members, where TGFβ exists in three different isoforms, TGFβ1, 2, and 3 (*Gilbert, Vickaryous & Viloria-Petit, 2016*). TGFβ1 plays a key role in bovine mammary gland involution by inducing apoptosis and autophagy in bovine mammary luminal epithelial cells (*Kolek et al., 2003*; *Gajewska, Gajkowska & Motyl, 2005*; *Zarzyńska, Gajewska & Motyl, 2005*). Our review of the bovine literature did not yield any reports of TGFβ1's effects on myoepithelial cells, specifically. Unlike some other species, alveolar structures remain intact and secretory epithelial and myoepithelial cells remain in the same position throughout bovine mammary involution and the dry period (*Holst, Hurley & Nelson, 1987*). Morphologically, bovine involution is characterized by reduced alveolar lumen area (*Holst, Hurley & Nelson, 1987*), as well as reduced integrity and increased permeability of tight junctions (*Pai & Horseman, 2011*). Polarized human and murine mammary luminal epithelial cells with well-established tight and adherens junctions as well as hemidesmosomes are resistant to apoptosis (*Weaver et al., 2002*; *Avery-Cooper et al., 2014*), and this might be the case for bovine mammary epithelial cells, as supported by a study by *Singh et al. (2005)* demonstrating that cell-extracellular matrix (ECM) communication is important in regulating bovine mammary epithelial cell survival. In other species such as mice, TGFβ1 is reported to dysregulate tight junctions causing mammary epithelial cells to undergo apoptosis (*Avery-Cooper et al., 2014*). Whether TGFβ1 disrupts tight junctions in the bovine mammary gland, and whether this is responsible for TGFβ1-induced apoptosis in the involuting bovine mammary gland, remains to be addressed.

Transforming growth factor beta 1 and its receptors are upregulated during bovine involution (*Plath et al., 1997*), and in vitro findings suggest that elevated TGFβ1 expression

during involution is due in part to its increased secretion by bovine mammary cells during the transition from lactation to the dry period (*Zarzyńska, Gajewska & Motyl, 2005*), as well as through enhanced release of TGFβ1 from the ECM (*De Vries et al., 2011*). TGFβ1 arrests bovine epithelial cell growth (*Kolek et al., 2003*) and such effect of TGFβ1 was shown to be important for the development of fully functional mammary glands in mice. In particular, TGFβ1 inhibited ductal branching and lateral extension via induction of Wnt5a expression and release (*Roarty & Serra, 2007*), and thus is currently considered a major regulator of ductal patterning in the mammary gland. It is unclear whether other isoforms, in addition to TGFβ1, are involved in lactation and/or involution in the bovine mammary gland. In this review, we focus specifically on the TGFβ1 isoform because it is the most commonly studied in the literature, but it is important to note that all three isoforms are present and differentially expressed in the bovine mammary gland (*Maier et al., 1991*). TGFβ1 and TGFβ3 are expressed by both the alveolar epithelium and the subepithelial stroma, while TGFβ2 is only expressed in epithelial cells (*Maier et al., 1991*). Bovine studies involving TGFβ3 should be encouraged in the future, given its demonstrated role in the involuting murine mammary gland, including repression of mammary luminal epithelial cell differentiation via induction of apoptosis (*Nguyen & Pollard, 2000*), and clearance of the dead epithelial cells via induction of phagocytic activity in the neighboring epithelial cell survivors (*Fornetti et al., 2016*).

In recent years, there has been a greater research focus on the mammary stroma and its influence on epithelial function. The stroma primarily consists of adipocytes and fibroblasts, but also contains endothelial cells, ECM, and inflammatory cells (*Kass et al., 2007*). TGFβ1 is a potent mediator of ECM synthesis and protects the ECM from degradation (*Lasky & Brody, 2000*; *Woodward et al., 2005*). Moderate amounts of fibrous tissue is a normal component of the involuting bovine mammary gland, and plays a major role in remodeling and in preparing the gland for the subsequent lactation period (*De Vries et al., 2010*). However, excessive TGFβ1 correlates with extensive tissue fibrosis, and this can interfere with the normal function of the bovine mammary gland, namely milk production (*Andreotti et al., 2014*). Excessive tissue fibrosis characterizes bovine mastitis (*Andreotti et al., 2014*), which is defined as inflammation of the mammary gland, and this is a major cause of financial losses to the dairy industry (*Heikkilä, Nousiainen & Pyörälä, 2012*). Furthermore, TGFβ1 is overexpressed in mastitic mammary glands in parallel with increased fibrosis and apoptosis of epithelial cells (*Andreotti et al., 2014*), and has been shown to enhance adhesion and invasion of *Staphylococcus aureus* to bovine mammary fibroblasts (*Zhao et al., 2017*), which makes TGFβ an attractive target of future research aimed to understand and control bovine mastitis. Further, with the strong evidence supporting TGFβ1's effects on both the stromal and parenchymal compartments of the mammary gland, a more holistic approach to studying TGFβ1 signaling, incorporating both stromal and epithelial cells, may be necessary to assess novel approaches for increasing milk production in a organ-like environment. In vitro treatment of whole tissue explants (*De Vries et al., 2011*; *Magro et al., 2017*) is an attractive possibility, as they may convey key mechanistic information impossible to obtain by analysis of biopsies. Another alternative is the use of three-dimensional co-culture models that
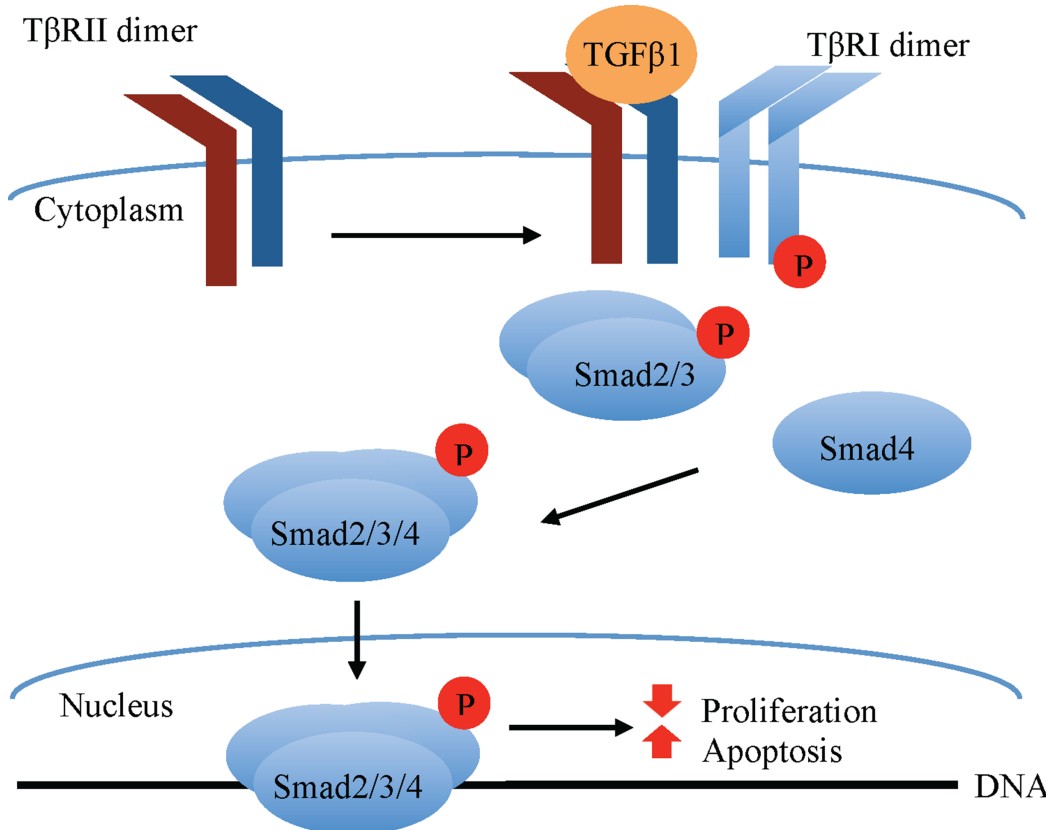

**Figure 1 Canonical TGFβ signaling.** TGFβ1 binding to the constitutively active TβRII Ser/Thr kinase promotes its re-localization and the formation of a tetrameric complex with the TβRI, leading to phosphorylation and activation of TβRI ser/thr kinase. The latter in turn phosphorylates the Smad2/3 transcription factors, permitting their association with Smad4. The Smad2/3/4 complex translocates to the nucleus where in association with other transcription factors (not shown in the figure) modulates the expression of target genes that, among other effects, promote apoptosis and inhibit cell proliferation in normal mammary epithelial cells.                

incorporate ECM and stromal cells of interest in addition to the epithelial cells, as those recently developed by our group (*Pallegar et al., 2018*).

## TGFβ1 signaling in the bovine mammary gland

Transforming growth factor beta 1 classically signals via a receptor serine/threonine kinase hetero-tetramer, comprised of equal parts TGFβ receptors I (TβRI) and II (TβRII). TGFβ1 ligands have high affinity for the type II but not type I TGFβ receptors (*Massagué, 1998*). Upon TGFβ1 ligand binding, the constitutively active TβRII dimer binds and phosphorylates TβRI, which becomes activated and phosphorylates receptor-associated small mothers against decapentaplegic (R-Smad) transcription factors, specifically Smad2 and Smad3. These R-Smads form a complex with the co-Smad, Smad4, which translocates to the nucleus, where it associates with other transcriptional elements to regulate gene transcription (Fig. 1). Among the factors affecting the outcome of canonical (Smad-mediated) TGFβ1 signaling are the type of R-Smad activated, the nature of the interacting transcriptional co-activators and co-repressors, as well as the

phosphorylation site, whether at the carboxy terminus or the central linker region (*Gilbert, Vickaryous & Viloria-Petit, 2016*).

In bovine mammary epithelial cells, TGFβ1 is documented to induce apoptosis and cell growth arrest through canonical Smad signaling (*Kolek et al., 2003*). In non-bovine cells, TGFβ1-induced apoptosis has also been reported to occur through non-canonical pathways such as mitogen-activated protein kinase (MAPK)/Erk, p38, c-Jun N-Terminal Kinase, PI3K/Akt, and Par6 signaling pathways (reviewed by *Zhang, 2009*; *Avery-Cooper et al., 2014*). As mentioned above, TGFβ1 has also been shown to inhibit mammary ductal branching in mice via Wnt signaling activation (*Roarty & Serra, 2007*). Not all of these pathways have been explored in bovine mammary epithelial cell apoptosis; however, downregulation of the PI3K/Akt and the MAPK/Erk pathways occurs in parallel with TGFβ1-induced apoptosis and growth arrest of bovine mammary epithelial cells in vitro (*Gajewska & Motyl, 2004*; *Di et al., 2012*).

Contrary to its inhibitory effect on the bovine mammary epithelium, TGFβ1 promotes stromal development in the mammary gland (*Musters et al., 2004*). TGFβ1 enhances bovine fibroblast proliferation through the MAPK/Erk pathway (*Gao et al., 2016*), and promotes the transition of fibroblasts to myofibroblasts (*De Vries et al., 2011*); the latter are key mediators of ECM protein synthesis and tissue fibrosis (*Phan, 2008*). Furthermore, TGFβ1 is known to cause bovine mammary epithelial cells to switch to a mesenchymal phenotype through a process known as epithelial-to-mesenchymal transition (EMT) (*Chen et al., 2017*). In bovine mammary epithelial cells, EMT was reported to occur (at least in part) through the Smad pathway (*Chen et al., 2017*). In non-bovine epithelial cells, other Smad-independent pathways have been implicated in TGF-β-induced EMT such as the MAPK/Erk pathway, p38 MAPK, c-Jun N-terminal kinases, Par6 signaling, and the PI3K/Akt pathway (*Xu, Lamouille & Derynck, 2009*; *Avery-Cooper et al., 2014*). A study exploring the use of tamoxifen as an inhibitor of tissue fibrosis, recently demonstrated that TGFβ induces a mammary fibroblast to myofibroblast transition via activation of MAPK/Erk signaling (*Carthy et al., 2015*). There is no current evidence on the involvement of specific non-canonical signaling pathways in TGFβ1-induced EMT or fibroblast to myofibroblasts conversion in bovine cells. Further research is necessary to understand the contribution of non-canonical signaling pathways to TGFβ1-induced apoptosis, EMT, activation of fibroblast into myofibroblasts, growth arrest, and mammogenesis in the bovine gland.

## TGFβ1 impact on apoptosis and growth arrest

One of the first studies to investigate TGFβ signaling in bovine mammary epithelium assessed the effect of serum starvation on TGFβ1 synthesis and secretion. *Zarzyńska, Gajewska & Motyl (2005)* mimicked the withdrawal of lactogenic hormones, growth factors, and nutrients that occurs at the end of lactation and the beginning of involution by reducing fetal bovine serum (FBS) content in the growth media from 10% to 0.5%, and analyzed TGFβ1 expression (*Zarzyńska, Gajewska & Motyl, 2005*). In media containing 0.5% FBS but not 10% FBS, TGFβ1 protein and mRNA expression was

increased (measured by laser scanning cytometry and RT-PCR, respectively), and reached its maximum levels after 24 h in BME-UV1 cells or 48 h in MAC-T cells. These results suggested that a factor in FBS suppresses TGFβ1 expression. The authors demonstrated that IGF-I (a key player in the somatotropic axis mediated by GH) suppresses TGFβ1 expression in BME-UV1 cells and MAC-T cells (*Zarzyńska, Gajewska & Motyl, 2005*). The percentage of apoptotic cells was also measured (using the sub-G1 region of a DNA histogram), and this reflected the patterns of TGFβ1 expression in MAC-T and BME-UV1 cells (*Zarzyńska, Gajewska & Motyl, 2005*). The investigators noted that there was a positive relationship between endogenous TGFβ1 expression and apoptosis, which provided support for the hypothesis that endogenous TGFβ1 induces apoptosis in these two cell lines. BME-UV1 cells were also reported to undergo apoptosis and growth arrest in response to exogenous (2 ng/mL) TGFβ1, and this was found to occur mainly through the intrinsic pathway of apoptosis (*Kolek et al., 2003*). *Gajewska, Gajkowska & Motyl (2005)* demonstrated the formation of a Smad-DNA complex in the nucleus 2 h after TGFβ1 treatment in BME-UV1 cells, indicating that the effects of TGFβ1 are transcriptionally mediated.

In 1995, *Woodward et al. (1995)* studied the influence of TGFβ1 on cell proliferation and cell death of MAC-T cells. This study found that maximal reduction in proliferation, as determined by total DNA measurements and thymidine incorporation, was obtained at 40 pM of TGFβ1. However, cytotoxicity (cell death) was not demonstrated by the trypan blue exclusion method using a TGFβ1 concentration as high as 40 nM (1,000-fold greater than 40 pM) (*Woodward et al., 1995*). This result contrasts the idea that TGFβ1 induces apoptosis of MAC-T cells, suggested by the *Zarzyńska, Gajewska & Motyl (2005)* study. The latter found that maximum levels of apoptosis paralleled maximum levels of TGFβ1 ligand after 48 h of serum starvation in MAC-T cells (*Zarzyńska, Gajewska & Motyl, 2005*) but it did not demonstrated that TGFβ1 actually caused apoptosis. Thus, the evidence of TGFβ1-induced apoptosis of MAC-T cells is weak if not inexistent, compared to that for BME-UV1 cells.

In addition to the studies discussed above with immortalized bovine MEC lines, an ex vivo study on bovine mammary gland explants revealed increased caspase 3 and cleaved Parp in parallel with increased levels of TGFβ1 and TβR2 during the dry period (*Zarzyńska et al., 2007*). These results provided indirect evidence that TGFβ1 induces apoptosis of bovine mammary epithelial cells in vivo. Furthermore, the group of *Di et al. (2012)* demonstrated that exogenous TGFβ1 induces apoptosis of primary bovine mammary epithelial cells isolated from the bovine mammary gland. This study revealed that levels of the death receptor ligand Fas ligand were reduced in parallel with increased levels of TGFβ1-induced apoptosis, which supports the findings by *Kolek et al. (2003)* demonstrating that TGFβ1-induced apoptosis in bovine mammary epithelial cells occurs through the intrinsic mitochondrial pathway. The results of this ex vivo assay demonstrate that primary bovine mammary epithelial cells undergo TGFβ1-induced apoptosis, similar to the BME-UV1 cells (*Kolek et al., 2003*), and this provides strong support for the idea that BME-UV1 mimic the behavior of cells directly isolated from the bovine mammary gland.

**Table 1 Effect of hormones on TGFβ1 expression in BME-UV1 and MAC-T cells.**

| Hormone | Cell lines | |
|---|---|---|
| | **MAC-T** | **BME-UV1** |
| Growth hormone | No significant effect | ↓ TGFβ1 protein levels in 10% FBS but ↑ TGFβ1 protein levels in 0.5% FBS |
| Somatostatin | No significant effect | ↑ TGFβ1 protein levels in 10% FBS |

**Note:**
Table summarizes select data from *Zarzyńska, Gajewska & Motyl (2005)*.

## Crosstalk between TGFβ1 and the somatotropic axis

*Gajewska & Motyl (2004)* found that suppression of PI3K/Akt signaling pathway was involved in the apoptotic response to TGFβ1 in BME-UV1 cells. In these cells, TGFβ1 was found to increase the expression of insulin-like growth factor binding protein (IGFBP)-3 and IGFBP-4, which led to the sequestration of IGF-I from the IGF-I receptor, reduction of PI3K/Akt signaling, and increased expression of the pro-apoptotic protein Bcl-2-associated death promoter (Bad) (*Gajewska & Motyl, 2004*). A separate study found that exogenous IGF-I administration to BME-UV1 cells completely blocked TGFβ1 expression and apoptosis, and this effect was reversed through the use of a PI3K inhibitor (*Zarzyńska & Motyl, 2005*). In agreement with the studies performed on BME-UV1 cells, *Di et al. (2012)* demonstrated that increased levels of Bad mediated the apoptotic response to TGFβ1 in primary non-immortalized bovine mammary epithelial cells isolated from the bovine mammary gland, which lends support to the hypothesis that TGFβ1-induced apoptosis in bovine mammary epithelial cells is mediated in part by a reduction of PI3K/Akt signaling. Altogether, this provides additional support for the argument that BME-UV1 cells closely resemble the responses to TGFβ1 that are seen in primary bovine mammary epithelial cells.

*Zarzyńska, Gajewska & Motyl (2005)* compared the effects of GH and somatostatin (a negative regulator of GH) on TGFβ1 expression in MAC-T and BME-UV1 cells (Table 1). GH significantly reduced TGFβ1 levels in BME-UV1 cells supplemented with 10% FBS (nutrient-rich conditions), while it significantly increased TGFβ1 levels in BME-UV1 cells supplemented with 0.5% FBS (nutrient-poor conditions). Treatment with somatostatin, a negative regulator of GH, was only evaluated in BME-UV1 cells supplemented with 10% FBS, in which it increased TGFβ1 protein levels. This suggests that in nutrient-rich conditions such as those expected during lactation, GH suppresses TGFβ1 expression in the bovine mammary gland, an effect that is blocked by somatostatin. In MAC-T cells, GH and somatostatin did not affect TGFβ1 protein levels (*Zarzyńska, Gajewska & Motyl, 2005*). Thus, TGFβ1 expression is altered by hormones of the somatotropic axis such as GH and somatostatin in BME-UV1 cells, but not in MAC-T cells (*Zarzyńska, Gajewska & Motyl, 2005*). Taken together with the demonstrated role of the somatotropic axis in the control of survival in bovine mammary epithelial cells (*Gajewska & Motyl, 2004*), the results discussed above suggest that the modulation of TGFβ1 levels and subsequent activity by the somatotropic axis is a mechanism to control survival/apoptosis in BME-UV1 cells but not in MAC-T cells.

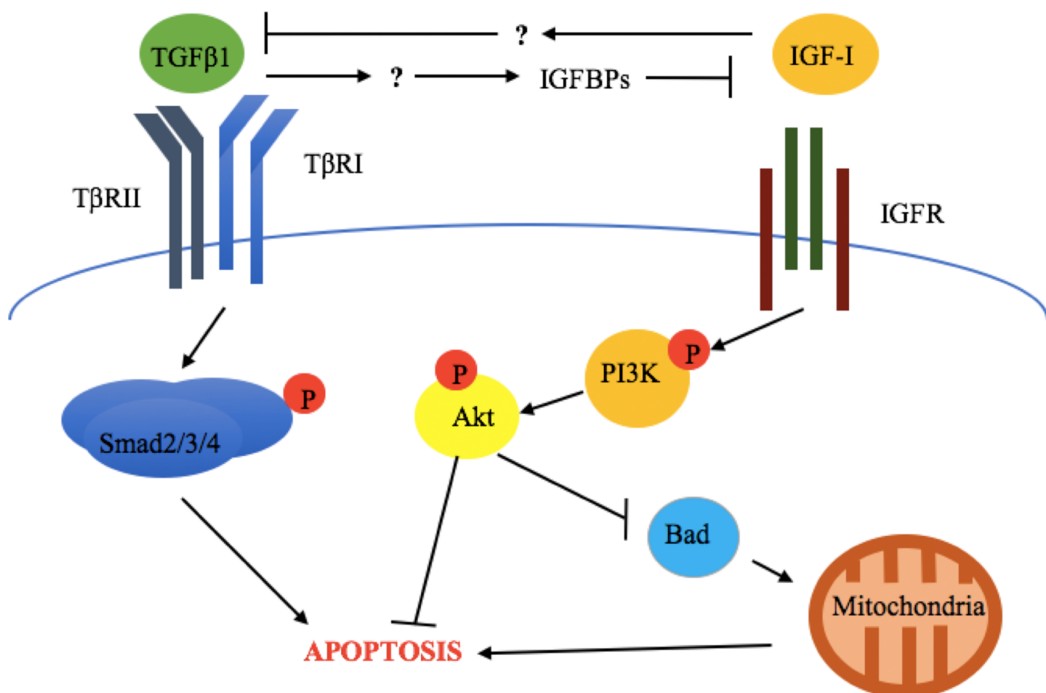

**Figure 2 Crosstalk between TGFβ1 and the somatotropic pathway.** TGFβ1 and IGF-1 repress each other' signaling and consequent cellular response (apoptosis and survival, respectively), in bovine mammary epithelial cells. This involves the inhibition of TGFβ1 expression by IGF-I and the blockade of IGF-I signaling activation by TGFβ1-induced IGF binding proteins (IGFBPs). The exact mechanisms by which these occur have not been described for bovine mammary epithelium.

The above results are also supported by in vivo evidence. *Zarzyńska et al. (2007)* evaluated the expression of TGFβ1, TGFβRII, IGF-I receptor α (IGF-IRα), IGF-I receptor β (IGF-IRβ), GH-R, IGFBP-3, -4, and -5, as well as biochemical markers of apoptosis (cleaved Parp and caspase 3) in bovine mammary gland explants at early lactation, late lactation, and during dry off (*Zarzyńska et al., 2007*). This study revealed that increased apoptosis, TGFβ1, TGFβRII, IGFBP-4, and-5 expression was accompanied by downregulation of GH-R and IGF-IRα during dry off (*Zarzyńska et al., 2007*). This suggests that TGFβ1 exerts its pro-apoptotic effects through suppression of the somatotropic pathway (Fig. 2).

As discussed above, IGF-I somatogenic effects have been shown to be caused, at least in part, by inhibition of TGFβ1 signaling (*Zarzyńska & Motyl, 2005*). Thus, IGF-I administration might be an effective way to block TGFβ1 signaling and its pro-apoptogenic and antiproliferative effects in vivo, an approach that has already been tested in an in vitro study by *Zarzyńska & Motyl (2005)* discussed above. However, in vivo studies examining the effect of IGF-I on milk production have yielded inconsistent results. *Davis et al. (1989)* compared the effects of administering systemic infusions of IGF-I or GH to lactating goats and measured the effects on milk production over a 10-day period. After 36 h, goats receiving GH had a milk yield increased by an average of 24%, while goats receiving IGF-I had no significant increase in milk yield compared to

saline-infused goats (*Davis et al., 1989*). In agreement with these results, intramammary infusion of IGF-I had no significant effect on milk protein expression or milk yield in dairy cows after a 4-day treatment interval (*Mackle, Dwyer & Bauman, 2000*). Conversely, infusion of IGF-I for 6 h via an external pudic arterial catheter in lactating goats resulted in milk yield significantly increased by 9% 2 h post-infusion, and this was associated with significantly increased mammary blood flow by 50–80% (*Prosser et al., 1994*).

Taken together, the above evidence does not consistently support an impact of IGF-I administration (and possibly indirect TGFβ1 inhibition) on acute milk yield following IGF-I administration. One limitation of these studies is that only short periods of time were used to examine milk production, rather than measuring the amount of milk produced over the entire lactation period, which has an average length of 225 days in goats and 305 days in cattle (*Ahuya et al., 2009*; *Bachman & Schairer, 2003*). It is possible that IGF-I administration could delay the onset of apoptosis and growth arrest that occurs in secretory epithelial cells and leads to the decline in lactation and the beginning of involution, without increasing peak milk yield. We propose that exogenous administration of IGF-I would extend the time in which peak yield is maintained compared to control cows. This area of research has not been recently investigated and many unanswered questions still remain regarding the impact of harnessing TGFβ1 signaling for the purposes of extending lactation. For instance, many studies have focused on TGFβ1's effect on cell number, but have not evaluated TGFβ1's possible effect on metabolic activities of secretory epithelial cells during lactation. This should be addressed in future studies.

## Phenotypic differences in MAC-T vs. BME-UV1 cells

In the previous section, we provided evidence of TGFβ1-induced apoptosis in BME-UV1, but not MAC-T cells (*Kolek et al., 2003*; *Woodward et al., 1995*), and we also discussed how the somatotropic axis controls TGFβ1 expression in the former but not the latter (*Zarzyńska, Gajewska & Motyl, 2005*). Comparing the literature on MAC-T and BME-UV1 cells yielded some very important differences that should be taken into account before using these cell lines to study TGFβ1 signaling.

The cell populations that compose the MAC-T and BME-UV1 cell lines were previously identified via determination of cell surface marker and cytokeratin (CK) expression under standard adherent, ultralow adherence, and three-dimensional (3D) cultures in laminin-rich ECM (*Arévalo Turrubiarte et al., 2016*). Staining of the luminal-specific CK CK19 and myoepithelial-specific CK14 (both human-specific antibodies) (*Borena et al., 2013*) was used to identify luminal and myoepithelial cell populations, respectively. Staining of CK19 and CK14 was supplemented by evaluating epithelial cell adhesion molecule (EpCAM) expression using a mouse-specific antibody. EPCAM is a cell-surface marker more highly expressed in bovine luminal-alveolar progenitor cells compared to other mammary epithelial cell types (*Perruchot et al., 2016*). To further support the luminal cell classification, *Arévalo Turrubiarte et al. (2016)* examined aldehyde dehydrogenase (ALDH) activity under standard adherence conditions, as ALDH is highly expressed in bovine luminal progenitor cells compared to other bovine mammary epithelial cell populations (*Martignani et al., 2010*).

**Table 2 Comparison of MAC-T and BME-UV1 luminal and basal marker expression in relation to one another.**

|  | MAC-T | BME-UV1 |
|---|---|---|
| CD10 expression | Absent | Present |
| CD49f expression | Present (higher in ADH) | Present |
| EpCAM expression | Present (lower in ADH and ULA) | Present |
| CK14 expression | Present | Absent |
| CK19 expression | Absent | Present |
| ALDH activity (ADH) | Lower | Higher |

**Note:**
Table summarizes select data from *Arévalo Turrubiarte et al. (2016)*. Unless otherwise indicated, the expression indicated reflects results obtained under three different culture conditions: standard adhesion (ADH), ultralow adherence (ULA), and 3D culture on laminin-rich ECM (Matrigel).

In order to assess basal phenotype, Areévalo Turrubiarte et al. assessed cluster of differentiation (CD) markers: CD49f (integrin α6 chain) using a rat-specific antibody and CD10 (neutral endopeptidase) using a mouse-specific antibody. CD49f is responsible for cell-matrix adhesions and for transmitting signals between mammary epithelial cells and the ECM (*Giancotti & Ruoslahti, 1999*). CD49f is expressed in cells of both luminal and basal origin; however, it is more highly expressed in basal cells (*Finot, Chanat & Dessauge, 2018*; *Rauner & Barash, 2012*; *Perruchot et al., 2016*). CD10, on the other hand, is normally used as a marker of basal cells (*Safayi et al., 2012*), but it is also enriched in mammosphere-forming cell populations; hence CD10 is also used as a marker of stem cells (*Maguer-Satta, Besançon & Bachelard-Cascales, 2011*). The ambiguity surrounding the significance of CD10 staining is addressed by staining for multiple markers in order to not misidentify cell phenotypes. For example, an additional marker of basal cells that can be used is CK14 (*Borena et al., 2013*), which was also included in the panel by the aforementioned researchers (*Arévalo Turrubiarte et al., 2016*). *Arévalo Turrubiarte et al. (2016)* marker expression assessment in BME-UV1 cells led to the conclusion that they have a luminal phenotype based on their high expression of EpCAM and CK19, and their high ALDH activity. Interestingly, the BME-UV1 cells also express CD10. In the absence of any other basal markers, these investigators interpreted the high CD10 expression and high ALDH activity of BME-UV1 cells as having a greater "stem-like" phenotype compared to the MAC-T cells (*Arévalo Turrubiarte et al., 2016*; Table 2).

The MAC-T cells were found by the same investigators to have a higher CD49f expression and lower ALDH activity compared to the BME-UV1 cells under standard adhesion culture conditions. In addition, MAC-T cells expressed CK14, a CK characteristic of myoepithelial cells, while the BME-UV1 cells did not (*Arévalo Turrubiarte et al., 2016*). The high expression of CD49f, the low ALDH activity, and the expression of CK14 suggest that MAC-T cells contain a myoepithelial cell population (*Arévalo Turrubiarte et al., 2016*; Table 2). Although the MAC-T cells contain characteristics indicative of myoepithelial cells, these cells were previously reported to lack oxytocin responsiveness and smooth muscle actin expression (*Zavizion, Gorewit & Politis, 1995*), which are two hallmarks of myoepithelial cells (*Gudjonsson et al., 2005*).

Further, *Arévalo Turrubiarte et al. (2016)* found MAC-T cells to lack CD10, which is a marker also expressed by mature bovine myoepithelial cells (*Perruchot et al., 2016*). It is important to note that the assessment of oxytocin and smooth muscle actin in MAC-T was performed following their culture as monolayers on glass (*Zavizion, Gorewit & Politis, 1995*). Instead, a 3D culture model on reconstituted basement membrane, such as Matrigel™, could be employed in order to examine the myoepithelial potential of the MAC-T cells. We based this suggestion in studies by *Mroue et al. (2015)*, in which a role for the gap junction protein Connexin 43 was demonstrated in the contractile response of myoepithelial cells to oxytocin via assessment of mouse derived mammary epithelial cell organoids under 3D culture conditions on Matrigel™. Appropriate ECM interactions under these conditions might promote differentiation into a myoepithelial phenotype capable of proper assembly of functional gap junctions and consequently contractibility following oxytocin exposure. Another thing to note about the study by *Arévalo Turrubiarte et al. (2016)* is the use of non-bovine specific antibodies (human-specific CK14 and CK19, rat-specific EpCAM, and mouse-specific CD49f and CD10), which does not take into account interspecies differences. To illustrate this concern, *The UniProt Consortium (2017)* revealed that human and bovine CK14 and CK19 have 91.1% and 88.8% protein sequence homology, respectively, while rat and bovine EpCAM and CD49f share 78.1% and 91.9% protein homology, respectively. Lastly, mouse and bovine CD10 share 91.3% protein homology (*The UniProt Consortium, 2017*). These differences in protein sequences between bovine, and the species against which the antibody was directed, can lead to misleading results not necessarily reflecting the true levels of expression or a given protein.

Another interesting feature of the MAC-T cells is their capability of synthetizing α- and β-casein, which is indicative that there are luminal/alveolar cells present within this cell line (*Huynh, Robitaille & Turner, 1991*). One explanation to the bi-phenotypic features of MAC-T cells, is that they contain bi-potent progenitor cells that gave rise to cells of both basal and luminal origin (*Rauner & Barash, 2012*). One study, which investigated the ability of MAC-T cells to form a functional mammary gland in vivo, suggests that this might be the case. MAC-T cells were mixed with the reconstituted ECM Matrigel™, and were implanted into the dorsal tissue of 8-week-old BALB/C nude male mice (*Park et al., 2016*). After 6 weeks, the transplanted tissue of these mice was dissected and analyzed for bovine mammary protein expression (*Park et al., 2016*). At this time, the MAC-T transplants had the characteristic alveolar structures of a female mammary gland (*Park et al., 2016*). These investigators evaluated the expression of CK18 and CK14 (luminal and myoepithelial CKs, respectively), both in the mammary tissue sections, as well as in vitro. The tissue sections expressed CK18, mainly localized in the luminal cells, and CK14, which was mainly found in the ductal and myoepithelial cells. The MAC-T cell line grown in monolayer culture stained positively for both CK14 and CK18 (*Park et al., 2016*). These results suggest that the MAC-T cell line contains a heterogeneous population. The above findings are additionally supported by a previous study in which MAC-T cells were subcloned into three distinct cell lines (*Zavizion, Gorewit & Politis, 1995*).

Both MAC-T and BME-UV1 cells are derived from lactating mammary epithelium, however, there were differences in the generation of these cell lines that should be considered. Although both cell lines were immortalized via transfection with simian virus 40 large T antigen (the temperature sensitive tsA58 mutant T antigen, specifically) and they were generated from mammary tissue obtained at slaughter from Holstein cows; BME-UV1 were obtained from a pregnant lactating cow (*Zavizion et al., 1996*), while MAC-T cells were obtained from a non-pregnant lactating cow (*Huynh, Robitaille & Turner, 1991*). Further, the transfection to generate BME-UV1 cells was carried out on a homogenous population of cells generated by subcloning a luminal-enriched cell population (*Zavizion et al., 1996*), while the transfection to generate MAC-T cells was performed on cells of epithelial morphology derived from serial dilution cloning of the original heterogeneous population of mammary cells (*Huynh, Robitaille & Turner, 1991*). Thus, differences between the MAC-T and BME-UV1 cells may be a consequence of the differing physiological statuses of the animals that were sampled and/or the cell population that was immortalized. It is also possible that the cells acquired different phenotypes due to lab-specific culture conditions since different culture conditions were used to generate the cells (*Arévalo Turrubiarte et al., 2016*). Such effect of culture conditions on the phenotype acquired by mammary epithelial cells has indeed been reported for human mammary cells (*Ince et al., 2007*).

## Summary and future directions

Using laser-scanning cytometry, *Zarzyńska, Gajewska & Motyl (2005)* demonstrated that endogenous levels of TGFβ1 in the bovine mammary gland epithelial cell line MAC-T reached their maximum in parallel with apoptosis. Apart from this study, there is no other evidence supporting that TGFβ1 induces apoptosis in MAC-T cells. Future experiments could use TβRI/II kinase inhibitors and/or TGFβ1 blocking antibodies, in order to conclusively demonstrate that apoptosis of MAC-T cells under starving conditions in induced by TGFβ1. Suppression of PI3K/Akt activity was reported to mediate TGFβ1-induced apoptosis of BME-UV1 cells (*Gajewska & Motyl, 2004*); however, the role of PI3K/Akt in TGFβ1-induced apoptosis of MAC-T cells has never been explored. Individual and combination treatments of exogenous TGFβ1, a TβRI/II kinase inhibitor, and a PI3K/Akt inhibitor would reveal the contribution of the PI3K/Akt pathway to the antiproliferative and apoptogenic effect of TGFβ1 in MAC-T cells. From the available ex vivo studies demonstrating TGFβ1-induced apoptosis in primary bovine mammary epithelial cells (*Di et al., 2012*; *Zarzyńska et al., 2007*), and the similar capacity of BME-UV1 cells to undergo apoptosis in response to TGFβ (*Kolek et al., 2003*), we can infer that BME-UV1 cells resemble the in vivo bovine mammary epithelium in relation to apoptosis.

The literature suggests that MAC-T cells may be composed of a bi-potent progenitor population based on the presence of both luminal and basal phenotypes (*Arévalo Turrubiarte et al., 2016*; *Park et al., 2016*). It would first be necessary to prove this by performing a clonal assay on MAC-T cells to demonstrate these cells give rise to both luminal and basal cells. A further step would be to compare the effect of TGFβ1 on

MAC-T cells to that on bi-potent progenitors derived from bovine mammary explants, in order to evaluate if cell line immortalization is a confounding variable. In recent years, several different models of epithelial cell hierarchy in the bovine mammary gland have been proposed (*Rauner & Barash, 2012*; *Perruchot et al., 2016*; *Finot, Chanat & Dessauge, 2018*). A better understanding of lineage commitment to basal or luminal phenotypes may shed light on the different origins of MAC-T and BME-UV1 cells, and thus guide their appropriate use in studies aimed to elucidate the role of TGFβ in bovine mammary epithelium apoptosis and bovine mammary gland biology in general.

In relation to the inverse relationship between the somatotropic and the TGFβ signaling pathway, studies in the past have found that IGF-I administration to dairy animals resulted in no change in milk production (*Davis et al., 1989*; *Mackle, Dwyer & Bauman, 2000*). We suggest that these results may reflect the study design. When evaluating the efficacy of a treatment, it is important that we address lactation persistency in terms of peak milk production as well as what happens before and after, by measuring milk production over the entire lactation period. IGF-I may not necessarily increase peak milk production in dairy cows, but it may still reduce the decline in milk production that inevitably follows peak milk yield. Another approach to reducing TGFβ1's potent apoptogenic and anti-proliferative effects on the epithelium and stimulatory effects on the stroma is to modify the dairy cow's diet. For example, *Gao et al. (2016)* demonstrated that dairy cows fed a diet of corn stover had higher serum levels of TGFβ1 compared to cows fed a diet of alfalfa. Furthermore, mammary glands of cows fed the diets of corn stover and alfalfa were examined ex vivo, and the former demonstrated increased levels of TGFβ1 and increased levels of vimentin (a marker of mammary stroma). This suggests that we can indirectly affect TGFβ1 signaling through modifying nutrition of dairy cows. This may be the most cost-efficient and practical approach to modifying TGFβ1 signaling in order to increase milk production, rather than the administration of growth factors or hormones such as IGF-I and rBST, respectively.

An important concern to note is that inhibition of TGFβ1 signaling can result in undesired side effects such as delayed wound healing and chronic inflammation (*Herbertz et al., 2015*). Furthermore, TGFβ1 plays an important role in mammary gland remodeling during involution since it is likely responsible for replacing senescent secretory epithelial cells (*Di et al., 2012*; *Zarzyńska et al., 2007*), and for promoting stromal development (*Musters et al., 2004*). Therefore, inhibiting TGFβ1 completely to prevent involution and the dry period would not be well-advised. In fact, *De Vries et al. (2011)* suggested that, rather than inhibiting TGFβ, exogenous TGFβ1 could be used during the first week of the dry period to hasten remodeling, shorten the dry period, and thus maximize milk production in the subsequent lactation period. It would be interesting to test this hypothesis in vivo by administering TGFβ1 during early dry period and comparing the effects to inhibition of TGFβ1 during the same time window, or comparing the same contrasting approaches during the late dry period. The length of the treatment is an important factor to take into consideration when designing these experiments, given the potential enhancement of mastitis susceptibility by TGFβ. A shorter treatment time might be required, based on the above-discussed evidence that TGFβ1 enhances

*S. aureus* adhesion and invasion into bovine mammary fibroblasts (*Zhao et al., 2017*). Taking this into account, any studies administering TGFβ1 in dairy cows should report prevalence of mastitis in different treatment groups to ensure that TGFβ1 is not predisposing dairy cows to intramammary infections.

Most researchers studying involution and lactation focus on the mammary epithelium, but recent work has demonstrated that TGFβ1 equally affects the mammary stroma. Future research is necessary to understand how TGFβ1's effect on the stroma affects the epithelial compartment. This would be particularly useful in determining if stromal-epithelial interactions affect how closely MAC-T and BME-UV1 cells match the response of in vivo bovine mammary epithelial cells to TGFβ1. We can evaluate the effect of epithelial-stromal interactions on TGFβ1 signaling by employing bovine mammary gland explants (*De Vries et al., 2011*; *Magro et al., 2017*), or a 3D culture system with re-constituted ECM (such as Matrigel™) to more accurately mimic the in vivo mammary gland (*Lee et al., 2007*). Another option is to use a co-culture system containing stromal cells (fibroblasts, adipocytes, or both) in addition to the epithelial cells (*Zhang et al., 2002*) or in addition to the epithelial cells and the ECM (*Pallegar et al., 2018*). Taking this approach will allow scientists to gain a better understanding of the interaction between epithelial and stromal compartments, and how this affects cell behavior.

## CONCLUSIONS

Transforming growth factor beta 1 signaling is a research interest in the field of mammalian lactation and involution, and can potentially be manipulated in order to increase lactation persistency. In the bovine mammary gland, MAC-T, and BME-UV1 cell lines have different responses to TGFβ1, and these responses should be carefully considered before employing them to study TGFβ1 signaling. BME-UV1 cells have been documented to undergo apoptosis in response to exogenous and endogenous TGFβ1, and this involved suppression of the somatotropic pathway (*Kolek et al., 2003*; *Zarzyńska, Gajewska & Motyl, 2005*; *Gajewska & Motyl, 2004*). Conversely, TGFβ1-induced apoptosis of MAC-T cells has never been demonstrated, and no involvement of the GH/IGF-I/PI3K/Akt pathway has been reported. Furthermore, hormones of the somatotropic pathway (GH and somatostatin) alter TGFβ1 expression in BME-UV1, but not in MAC-T cells (*Zarzyńska, Gajewska & Motyl, 2005*).

The different responses of BME-UV1 and MAC-T cells to TGFβ1 is possibly a result of their differing phenotypes. BME-UV1 cells express a luminal phenotype, while MAC-T cells possess a phenotype indicative of both luminal and myoepithelial populations (*Arévalo Turrubiarte et al., 2016*). For researchers wishing to study interactions between TGFβ1 signaling and the somatotropic pathway in alveolar bovine mammary epithelial cells, the current evidence suggests that BME-UV1 cells better reflect the physiology of milk-secreting bovine alveolar mammary epithelial cells.

Taking into account TGFβ1's effect on mammary stroma and the role of stromal-epithelial interactions in mammary gland physiology, the influence of stromal cells is an important factor to consider when testing how closely bovine mammary cell lines (including MAC-T and BME-UV1) resemble the in vivo mammary gland.

The experimental design for future research involving MAC-T and BME-UV1 cells should keep this in mind.

## ACKNOWLEDGEMENTS

We would like to recognize all the support and suggestions we received from members of the Viloria-Petit, Coomber, and Mutsaer laboratories, as well as from a number of other members of the Department of Biomedical Sciences at the University of Guelph.
Our special thanks to Charles Mitz and Mary Ellen Cybulski for their critical review of the manuscript. We dedicate this work to the memory of C.A.M. grandfather, Larry Mitz.

### Funding

This work was supported by a National Sciences and Engineering Research Council (NSERC) of Canada grant to Alicia Mercedes Viloria-Petit (RGPIN-2017-3977). The funders had no role in study design, data collection and analysis, decision to publish, or preparation of the manuscript.

### Grant Disclosure

The following grant information was disclosed by the authors:
National Sciences and Engineering Research Council (NSERC) of Canada grant to Alicia Mercedes Viloria-Petit: RGPIN-2017-3977.

### Competing Interests

The authors declare that they have no competing interests.

### Author Contributions

- Charlotte Alexandra Mitz conceived and designed the experiments, performed the experiments, analyzed the data, prepared figures and/or tables, authored or reviewed drafts of the paper, approved the final draft.
- Alicia Mercedes Viloria-Petit conceived and designed the experiments, performed the experiments, analyzed the data, prepared figures and/or tables, authored or reviewed drafts of the paper, approved the final draft.

### Data Availability

    This is a review article and did not generate raw data.

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
