# Peer review of "TGF-beta signalling in bovine mammary gland involution and a comparative assessment of MAC-T and BME-UV1 cells as in vitro models for its study"

_PeerJ, doi:10.7717/peerj.6210_

## Round 0.1 · original submission · Major Revisions

All reviewers agree the review is of interest, but all highlight additions/changes that could improve the relevance of the article. In particular, some discussion of the relevance of the cell lines to primary cells is important to address. In addition, reviewer 3 has made some clear suggestions that I would strongly encourage you to address. These changes represent relatively major revisions, but I think are clearly stated and hopefully straightforward to address. I look forward to receiving your revised manuscript.

·

Basic reporting

No comment.

Experimental design

Additional information should be added on how accurately the MAC-T and BME-UV1 cells mimic mammary gland cells assayed ex vivo from cattle.
Please include some information to place into context the suggestion that MAC-T are myoepithelial cells and BME-UV1 are luminal. Where are these cells placed in the intact mammary gland? Do they differ in lactating and non-lactating glands? How relevant are they to the process where TGFb1 may have an impact in vivo?

Validity of the findings

If no existing literature exists then a suggested question for the future would be to address how closely these mammary cell lines compare to those directly isolated or identified in the bovine mammary gland. This would provide very useful information as to whether they are useful to assess the utility of approaches such as TGFb1 inhibition for extended lactation/improved milk yield.

·

Basic reporting

no comment

Experimental design

no comments

Validity of the findings

no comments

Additional comments

This is a really interesting review regarding the impact of TGFb1 on two different bovine udder-derived cell line.
As this topic is probably not the most know to readers of Peerj, could the authors potentially re-structure the lay out by starting with
1) Describing phenotypic differences of Mac-T vs BME-UV1
2) Describe TGFb1 signalling in general
3) Then follow with the specifics

What this reviewer is missing a bit is the notion of whether this is exclusively specific for TGFb1, or may also be similar for TGFb2, is it important where the cell lines are coming from (as in lab sources of either lines used in the different articles cited), and whether the antibodies to CD mmarkers tested were bovine specific.

Reviewer 3 ·

Basic reporting

- Is the review of broad and cross-disciplinary interest and within the scope of the journal?
This mini review is not of broad and cross-disciplinary interest.

-Has the field been reviewed recently? If so, is there a good reason for this review (different point of view, accessible to a different audience, etc.)?
The field has not be reviewed recently, but the manuscript presents not enough up-to-date information on the subject described.

-Does the Introduction adequately introduce the subject and make it clear who the audience is/what the motivation is?
Yes.

Experimental design

-Is the Survey Methodology consistent with a comprehensive, unbiased coverage of the subject? If not, what is missing?
Yes.

- Are sources adequately cited? Quoted or paraphrased as appropriate?
Yes.

- Is the review organized logically into coherent paragraphs/subsections?
Yes.

Validity of the findings

- Is there a well developed and supported argument that meets the goals set out in the Introduction?
No,see general comments.
-Does the Conclusion identify unresolved questions / gaps / future directions?
No,see general comments.

Additional comments

The mini review (#29236) by Mitz Ch.A. and Viloria-Petit A.M. describes the apoptogenic and anti-proliferative effect of TGF-beta1 in two bovine mammary epithelial cell lines: MAC-T and BME-UV1, and is based mainly on data described in scientific literature in years 1995-2005, although the chapter regarding the phenotypic differences in MAC-T vs. BME-UV1 cells has been created using also more recent findings (2010-2016). The authors have undertaken this topic due to the continuous need of dairy industry to extend lactation persistency in cows, and thus search for possible new solutions to reach this goal. Such solutions should be alternative to exogenous administration of recombinant bovine growth hormone to dairy cows. The Authors hypothesize that inhibiting TGF-beta1 signaling pathway may be helpful in extending lactation and increasing milk production, due to the fact that this cytokine exerts apoptogenic and anti-proliferative actions in bovine MECs. This hypothesis I quite risky considering the fact that no in vivo studies have been conducted to test it, and it is impossible to predict what will happen if TGF-beta1 becomes inhibited by e.g. exogenous factors, considering the fact that this cytokine plays an important regulatory role in many different tissues.
Important issues decreasing the value of this review:
1. The studies commented in this mini review were done at the beginning of XXI century using commercially available, immortalized bovine MECs lines. There is no data on the effect of TGF-beta1 on primary bovine MECs, isolated directly from the udder, and thus maintaining more properties of mammary epithelium within the mammary gland. This should be taken under consideration by the authors and commented in this review.
2. Nowadays a growing body of evidence point at important interactions between the mesenchymal and stromal tissue of the mammary gland during its development and remodeling, and TGF-beta has been shown to play an important role also in regulation of the stromal compartment in bovine mammary gland (Musters S. et al., J Dairy Sci. 2004;87(4):896-904.; Andreotti CS, et al.,Res Vet Sci. 2014 Feb;96(1):5-14). This review would be more interesting if the authors presented more comprehensive approach discussing not only the available data of in vitro studies, but also studies on animals (in vivo studies on bovine mammary gland) to show pros and cons of using TGF-beta inhibition as a potentially new method of extending the lactation in cattle.
3. In addition, the review is lacking information about the noncanonical pathways induced by TGF-beta1, which may be of great importance. The reader of such review should be informed that aside from the classic, canonical Smad-mediated pathway, other signaling pathways, mediated by Wnt or MAP kinases may also be activated by this cytokine, and thus inhibition of TGF-beta1 activity may result in other consequences beside inhibiting MECs apoptosis.
Overall, the article is written in clear written in professional language, but in the current form it is lacking up-to-date findings regarding the role of TGF-beta1 in bovine mammary gland, thus it may be of limited interest.

---

## Round 0.2 · accepted · Accept

Your revisions have satisfied all three reviewers and I am very happy to accept the manuscript in its current format.

# ·

Basic reporting

No comment

Experimental design

No comment

Validity of the findings

No comment

Additional comments

The authors have addressed this reviewers concerns and comments.

·

Basic reporting

Revised version of previously submitted manuscript

Experimental design

Revised version of previously submitted manuscript

Validity of the findings

Revised version of previously submitted manuscript

Additional comments

The authors have answered all my comments raised in a very satisfactory manner

Reviewer 3 ·

Basic reporting

With reference to remarks in my first review:
- the manuscript has been largely improved in terms of its broad and cross-disciplinary interest. It is now within the scope of the journal.
- Furthermore, the manuscript was enriched by substantial amount of up-to-date information on the subject described.
- Introduction has been improved.

Experimental design

-Is the Survey Methodology consistent with a comprehensive, unbiased coverage of the subject? If not, what is missing?
Yes.

- Are sources adequately cited? Quoted or paraphrased as appropriate?
Yes.

- Is the review organized logically into coherent paragraphs/subsections?
Yes.

Validity of the findings

- Is there a well developed and supported argument that meets the goals set out in the Introduction?
Yes
-Does the Conclusion identify unresolved questions / gaps / future directions?
Yes

Additional comments

The present version of the manuscript by Mitz and co-workers has been substantially improved following the suggestions of the Reviewers. In my opinion this version is well structured, informative and easy to follow, therefore I recommend the manuscript for publication in Peer J journal.